# The role of feedforward and feedback inhibition in modulating theta-gamma cross-frequency interactions in neural circuits

**Dimitrios Chalkiadakis[1,2], Jaime Sánchez-Claros[2], Víctor J. López-Madrona[3], Santiago Canals[1]\*, Claudio R. Mirasso[2]\***

**1** Instituto de Neurociencias, Consejo Superior de Investigaciones Científicas, Universidad Miguel Hernández, Sant Joan d'Alacant, Spain, **2** Instituto de Física Interdisciplinar y Sistemas Complejos (IFISC, UIB-CSIC), Campus UIB, Palma de Mallorca, Spain, **3** Aix Marseille Univ, INSERM, INS, Inst Neurosci Syst, Marseille, France

\* scanals@umh.es (SC); claudio@ifisc.uib-csic.es (CRM)

## Abstract

Interactions among brain rhythms play a crucial role in organizing neuronal firing sequences during specific cognitive functions. In memory formation, the coupling between the phase of the theta rhythm and the amplitude of gamma oscillations has been extensively studied in the hippocampus. Prevailing perspectives suggest that the phase of the slower oscillation modulates the fast activity. However, recent metrics, such as Cross-Frequency Directionality (CFD), indicate that these electrophysiological interactions can be bidirectional. Using a computational model, we demonstrate that feedforward inhibition modeled by a theta-modulated ING (Interneuron Network Gamma) mechanism induces fast-to-slow interactions, while feedback inhibition through a theta-modulated PING (Pyramidal Interneuron Network Gamma) model drives slow-to-fast interactions. Importantly, in circuits combining both feedforward and feedback motifs, as commonly found experimentally, directionality is flexibly modulated by synaptic strength within biologically realistic ranges. A signature of this interaction is that fast-to-slow dominance in feedforward motifs is associated with gamma oscillations of higher frequency, and *vice versa*. Using previously acquired electrophysiological data from the hippocampus of rats freely navigating in a familiar environment or in a novel one, we show that CFD is dynamically regulated and linked to the frequency of the gamma band, as predicted by the model. Finally, the model attributes each theta-gamma interaction scheme, determined by the balance between feedforward and feedback inhibition, to distinct modes of information transmission and integration, adding computational flexibility. Our results offer a plausible neurobiological interpretation for cross-frequency directionality measurements associated with the activation of different underlying motifs that serve distinct computational needs.

**Data availability statement:** The electrophysiological datasets used in this study are available at: https://doi.org/10.20350/digitalCSIC/12537. Codes can be assessed at https://github.com/gerompampastrumf/thetaING_PING.

**Funding:** DC, JSC and CRM acknowledge support from the Spanish Ministerio de Ciencia, Innovación y Universidades through projects PID2021-128158NB-C22 and María de Maeztu CEX2021-001164-M funded by the MICIU/AEI/10.13039/501100011033. DC and JSC acknowledge support from the Spanish Ministerio de Ciencia, Innovación y Universidades through projects PID2021-128158NB-C21 and Severo Ochoa CEX2021-001165-S funded by the MICIU/AEI/10.13039/501100011033. The funders had no role in the study design, data collection and analysis, the decision to publish, or the preparation of the manuscript.

**Competing interests:** The authors have declared that no competing interests exist.

## Author summary

We investigate the interaction between various types of brain oscillations and their potential relationship with the connectivity of underlying neural networks. Brain activity encompasses slow oscillations, such as theta, alpha, and delta, as well as faster oscillations, including gamma. These oscillations interact through Cross-Frequency Coupling (CFC), a mechanism essential for cognitive processes like memory, learning, and attention. Given the higher spectral power and broader spatial propagation of slow oscillations, it has been proposed that CFC arises when slow oscillations modulate faster activity. However, recent evidence suggests that gamma oscillations can also predict the phase of slower oscillations, indicating a bidirectional relationship. To explore this complexity, we developed a computational model that reproduces both forms of interaction observed experimentally. Our results demonstrate that while slow oscillations originating from distant regions can induce gamma activity, local connectivity and specific cell-type dynamics allow gamma oscillations to anticipate slow oscillations. Importantly, directionality is modulated by the balance between feedforward and feedback inhibitory circuits, giving rise to distinct dynamical modes with specific computational properties that enhance the flexibility of the system. This work integrates competing hypotheses on oscillation interactions and offers a conceptual framework for linking dynamics to the structural organization of neural circuits.

## Introduction

Mammalian brains exhibit oscillatory activity over a broad frequency range, from 0.5 to 500 Hz, spatially organized across different regions [1]. These oscillations reflect distinct synchronization patterns of the underlying neural circuits and are associated with different behavioral states. For instance, theta band oscillations, around (4–8) Hz, in the prefrontal cortex and hippocampus and alpha band oscillations, around (8–12) Hz, in the visual cortex, have been linked with locomotion, learning, and attention [2–6].

Faster rhythms, such as gamma oscillations within the 30–150 Hz frequency band, are also ubiquitous in brain networks [7,8]. Notably, gamma oscillations frequently interact with slower rhythms in a phenomenon known as cross-frequency coupling (CFC) [9]. The hippocampus in particular, has been extensively studied for its theta-gamma CFC, which shows increased gamma amplitude locking to the phase of the theta cycle during decision-making and learning [10–12]. Additionally, CFC has been observed in the cortex, where gamma activity couples with theta, alpha, and beta oscillations [13]. Theories of neural computation propose that CFC plays a critical role in inter-regional communication essential for attention [14], and in organizing neuronal firing into cell assembly sequences underlying episodic memory formation [15].

The generation of CFC, though not fully understood, has been explained based on local and global network properties. *In vitro* and computational studies identified intrinsic neuronal properties, such as $I_h$ currents, and interactions between fast and slow interneurons, as key local mechanisms [16–20]. Simultaneous electrophysiological recordings in the hippocampus and entorhinal cortex showed, however, that CFC is also influenced by rhythmic inputs from upstream layers [21–23]. The finding of high coherence at low frequencies across distant recording sites, but not at high frequencies, led to the concept that slow oscillations are driven by upstream areas, which then locally organize faster network dynamics [24–26]. This aligns with the oscillatory hierarchy hypothesis, which posits that slower oscillations modulate population excitability, thereby coordinating higher-frequency activity [27].

The application of techniques for separating field potentials into pathway- or layer-specific activity patterns- [28], combined with new metrics that assess directional interactions across frequency bands, such as cross-frequency-directionality (CFD) [29], has revealed bidirectional interactions between fast and slow oscillations. In the hippocampus, recordings from rats engaged in navigation and memory tasks demonstrated that bouts of gamma activity systematically preceded the phase of theta oscillations, suggesting gamma-to-theta interaction [12]. Similarly, human electrocorticography studies have reported gamma-to-alpha interactions in the visual cortex [29,30]. Furthermore, using two independent methods to assess directionality, Dupré la Tour and colleagues [31] observed both interactions: theta-to-gamma in the hippocampus during REM sleep in rats, and gamma-to-theta in the human auditory cortex. Overall, CFC is not exclusively due to slow-to-fast interactions challenging the conventional oscillatory hierarchy hypothesis.

In this study, we aim to (1) investigate the emergence of directionality by employing a computational modeling approach and (2) offer a plausible neurobiological interpretation of CFD. Specifically, we adapt a model from the work of [32] to investigate the underlying mechanisms and provide new insights. In our model, pyramidal neurons receive inputs from an external population generating a slow (theta) rhythm, while the fast (gamma) rhythm emerges locally through the activity of fast-spiking interneurons. We found that, locally, the temporal relationship between theta and gamma and, consequently, the cross-frequency directionality, is determined by the dominance of specific connectivity motifs within the underlying circuit. In a theta-modulated Interneuron Network Gamma (θ-ING) motif, feedforward recruitment of fast-responding interneurons primarily drives gamma-to-theta directionality. In contrast, in theta-modulated Pyramidal-Interneuron Network Gamma (θ-PING) motif, feedback inhibition supports theta-to-gamma directionality. In combined motifs that reflect the anatomy of neuronal circuits commonly found in the brain, we analyzed transitions between these modes, demonstrating smooth bidirectional interactions controlled by synaptic strength within biologically plausible ranges. We validated several model predictions using previously acquired experimental data [12]. Finally, by evaluating the capacity of each computational motif to integrate distinct inputs, we uncovered a mechanism that prioritizes transmission across different parallel information channels arriving at the dendrites of pyramidal cells.

## Methods

### Theta-gamma generation in the θ-ING and θ-PING motifs

The two motifs analyzed in this study are variations of the Interneurons Network Gamma (ING) and Pyramidal-Interneuron Network Gamma (PING) models, both of which generate gamma activity through interactions between pyramidal cells (PCs) and fast-spiking inhibitory interneurons, primarily parvalbumin-immunoreactive basket cells (BCs) [33–37]. In both models, a gamma cycle begins with BCs firing, which suppresses the activity of their target neurons (both PCs and other BCs). Once the inhibition decays, BCs fire again, initiating a new cycle. The key difference between the two models lies in the source of excitatory drive to BCs: in the ING model, excitation originates from outside the network, whereas in the PING model, it results from bidirectional interactions between BCs and PCs.

We modified the traditional ING and PING models by incorporating an external theta drive that regulated the activation and suppression of gamma dynamics, thereby inducing theta-gamma cross-frequency coupling (CFC). The θ input was

modeled such that the interval between consecutive peak activations, $T_{cycle}$, was drawn from a Gaussian distribution with a mean $\mu_T = 125$ms and a standard deviation $\sigma_T = 16$ms. Each cycle contained 10,000 spikes, distributed according to a second Gaussian with a standard deviation $w = 25$ms. The variability in both inter- and intra-cycle timing, controlled by $\sigma_T$ and $w$, respectively, was essential to detect directional interaction between theta and gamma rhythms.

In order to make meaningful comparisons between the two motifs we used the same synaptic weights for the connections θ→PCs and BCs→PCs while the PCs→BCs synaptic strength of the θ-PING was chosen so that the mean firing rate of the PCs was almost identical in both motifs (see S1 Table for more details). The firing rates are $f_{r,\theta\text{-ING}} = 0.48(0.01)$ Hz and $f_{r,\theta\text{-PING}} = 0.49(0.01)$ Hz except if stated otherwise, where the mean value and the standard deviation (in parenthesis) were calculated across 20 simulations. Despite receiving the same, relatively weak theta input—selected to reproduce the sparse, in vivo–like firing where individual neurons spike at rates below the population rhythm (so-called weak PING/ING [38])—the global gamma oscillations in the θ-ING and θ-PING motifs exhibited markedly different frequencies. This divergence reflects the circuit architecture. In the θ-ING motif, fast-spiking BCs are driven directly by the theta rhythm and rapidly generate gamma oscillations. In contrast, in the θ-PING motif, PCs—with their slower membrane time constants— require more time before recruiting BCs, thereby yielding a lower γ-frequency.

## Cells

Although the motifs we analyzed are ubiquitous in multiple brain areas, our model was adapted from previous works modeling the hippocampus CA3 area in [32]. Pyramidal cells were represented as multi-compartmental neurons with active dendrites while basket cells were modeled as single-compartment point neurons. Both neuron types included leak, transient sodium, and delayed rectifier potassium currents, with PCs also incorporating hyperpolarization-activated currents. For a comprehensive description of the model see also chapter 4 section 4.2 of [39] and ref. [32]; the simulations code is available in GitHub (https://github.com/gerompampastrumf/thetaING_PING).

Modeling multicompartmental pyramidal cells with dendrites allows for the simulation of realistic transmembrane currents and Local Field Potential (LFP) recordings [40]. To achieve this, our PCs consisted of five compartments, each subdivided into three segments to enhance spatial integration accuracy. Of them, 3 compartments emulated the apical dendrites, one the soma and another one the basal dendrites. All compartments were modeled as cylinders.

For the BCs we chose not to model the dendritic tree explicitly but considered them as point neurons, following ref. [32]. However, since dendritic transmission time in BCs is small but not negligible, we incorporated an additional fixed delay of 1 ms from the external driver to the basket cells. This dendritic delay was selected to exceed the onset time between evoked dendritic postsynaptic currents and the corresponding increase in somatic membrane potential observed in patch-clamp experiments (see Fig 3d of [41]). For comparison, both our multicompartmental model and experimental results, suggest that the dendritic delays in PC are between 2–5 ms depending on the position of the stimulus and whether it evokes a dendritic spike or not [42,43].

## Synapses

Our model included three types of synaptic connections: inhibitory $GABA_A$, excitatory AMPA and NMDA, all of which were modeled using the standard double exponential function [32]. Synaptic weights were chosen so that firing rates reflect realistic PC and BC activity [22], with NMDA connections fixed at a value ten times lower than those for AMPA. Neuronal connections were chosen randomly. Axonal propagation and synaptic transmission delays between populations $x$ and $y$ were modelled using a transmission delay $\tau_{x,y}$, drawn from a Gaussian distribution with a standard deviation of 0.2 ms. The mean of the distributions were set as follows: $\tau_{\theta,BC} = 20$ ms, $\tau_{\theta,PC} = 20$ ms, $\tau_{BC,PC} = 1.5$ ms, $\tau_{PC,BC} = 1.5$ ms, and $\tau_{BC,BC} = 1.5$ ms. The relatively large values for $\tau_{\theta,BC}$ and $\tau_{\theta,PC}$ were chosen to ensure that the relative difference between the two could be manipulated without resulting in negative delays. To account for background activity, Poissonian noise sources with a rate of 1 ms were added to the pyramidal soma, proximal dendrites, and basket cells. We also examined the competition

between a parallel pathway, incorporated in the proximal dendrite, and the θ input. The parallel pathway was modeled as a Poissonian source with its synaptic strength increased tenfold with respect to the background activity and its rate adjusted from 1 ms to 5 ms. The values of the synaptic parameters are detailed in S1 and S2 Tables.

## Simulations

Simulations were conducted using the NEURON simulator library [44]. Each simulation modeled 60 seconds of activity across 20 realizations, with different realizations of the Poissonian and θ-inputs. An exception was made for simulations comparing the functional differences between the two motifs: depending on the protocol, a selected input was kept fixed across simulations, while different realizations were used for the remaining inputs. Each simulation included 200 PCs and 40 BCs.

## Dataset

All experimental data used in this study were previously acquired and described in [12]. Briefly, the dataset (available at https://doi.org/10.20350/digitalCSIC/12537) comprises of electrophysiological recordings (32-channel silicon probes) from five rats along the dorso-ventral axis of the dorsal hippocampus, covering the CA1 region and the dentate gyrus. During the recordings, the animals freely explored an open field to which they had been previously habituated over the course of one week through daily sessions (familiar environment condition). Subsequently, a modification was introduced to the open field—a change in the surface texture of the floor—and the animals were recorded again (mismatched novelty condition). The results of theta-gamma CFC and CFD under these conditions can be found in [12]. The specific dataset used in the present study consists of pathway-specific field potential from the lacunosum-moleculare layer of CA1, which exhibits the highest signal-to-noise ratio for theta and gamma activity. This field potential was obtained by applying Independent Component Analysis (ICA) to the 32 time series of local field potentials (LFP) recorded using the multichannel electrodes [12,28].

## Analysis

### Cross frequency coupling

This metric aims to quantify the degree to which the amplitude of a signal at a given high frequency $f_{ampl}$ co-modulates with the phase at a different low frequency $f_{phase}$. Various metrics have been developed for this purpose, but we will use the one proposed by Canolty et al. in [45], also known as the mean vector length. This method involves first filtering the signal at both $f_{ampl}$ and $f_{phase}$, followed by the calculation of the analytic signal's amplitude at high frequencies $A_{ampl}(t)$ and the analytic signal's phase at low frequencies $\varphi_{phase}(t)$. Next, we construct the composite signal $z(t) = A_{ampl}(t)\exp(i\varphi_{phase}(t))$, which resides in the complex plane. If there is no coupling between the selected frequencies, the trajectory of $z(t)$ will be radially symmetric. Consequently, the absolute average of the composite signal, $z_{avg}$, is zero when there is no CFC and positive otherwise. In this study, we used the Comodulogram class from the pactools Python library created by Dupré la Tour and colleagues [31] to extract the absolute value of $z_{avg}$.

To statistically assess the significance of the results, we employ a surrogate analysis followed by clustering correction for multiple comparisons [29]. We generate 1,000 surrogate time series by randomly splitting the phase signal into two segments and swapping their order. This procedure disrupts the temporal relationship between $\varphi_{low}(t)$ and $A_{high}(t)$ while preserving key characteristics such as their spectra. We then calculate the 99th percentile value, ($k_{th}$), which represents the CFC value greater than 99% of the samples across all surrogate data. All CFC values below $k_{th}$ are set to zero. Subsequently, clusters of adjacent non-zero CFC values are identified within each dataset, both surrogate and non-surrogate. Each cluster is assigned a cluster score calculated as the sum of the values within the cluster. The CFC of the original data is considered significant at $p < 0.01$ if its cluster score exceeds the 99th percentile of all surrogate cluster scores.

## Cross frequency directionality

While the mean vector length metric effectively detects co-modulation of slow and fast components within a signal, it does not provide insights into potential temporal relationships between them (for an illustrative example see S1 Fig). To address this limitation, we use the cross-frequency directionality (CFD) metric developed in [29]. CFD utilizes the phase slope index (PSI), which quantifies the directionality between two broadband signals, *x* and *y*, by analyzing the relationship of their phase difference, $\Delta\varphi = \varphi_x - \varphi_y$ as a function of frequency. When the phase differences are linearly dependent on frequency, it indicates a fixed time lag between the two signals. Specifically, if $\Delta\varphi$ increases with frequency, *x* leads *y*; conversely, if $\Delta\varphi$ decreases, *y* leads *x*.

In more detail, the PSI is calculated as follows. First, we compute the Fourier Transform of the signals *x* and *y* denoted *X* and *Y*, respectively. These series are then split into N smaller segments, and the complex coherence is computed:

$$C(f) = \frac{\sum_{i=1}^{N} X_i \cdot Y_i^*}{\sqrt{\sum_{i=1}^{N} |X_i|^2 \sum_{i=1}^{N} |Y_i|^2}},$$

(1)

where '*'denotes the complex conjugate. Then, the phase slope index is calculated as:

$$PSI(f_i) = Im\left(\sum_{j=f_i-\frac{\beta}{2}}^{j=f_i+\frac{\beta}{2}} C^*(f_j) C(f_j + \Delta f)\right),$$

(2)

where "Im" refers to imaginary part. In cross-frequency analysis, the PSI is adapted when *y* is not a separate signal from *x*, but rather its amplitude filtered around a high frequency, $f_{high}$. Finally, to emphasize the directionality of regions exhibiting strong cross-frequency coupling, we apply the masking technique described in [12]. Specifically, we divided the CFC by its maximum value so that it is bounded between 0 and 1, thus creating a mask. The final CFD is then calculated as the phase slope index multiplied by this mask.

To statistically assess the significance of the results, we employ the same procedure as for CFC for the highest 99th quantile. We also repeat the process for values lower than the 1st quantile in order to detect regions of significant negative CFD [29].

In certain cases, the predominant $CFD_{avg}$ is extracted from the two-dimensional comodulogram. To achieve this, we generate an alternative binary mask over the CFC, retaining only frequencies where CFC exceeds the 0.95 quantile. We then compute $CFD_{avg}$ by averaging all CFD values at these selected frequencies. Furthermore, we confirm that the results remain consistent when using alternative quantiles, such as 0.9 and 0.85.

## From spikes to continuous timeseries

As part of the analysis, we examined the instantaneous output of a neuronal population. To achieve this, discrete spike events were transformed into continuous time series. In S2 Fig, we illustrate the transformation where we convolve all spikes with two different kernels: a Gaussian kernel and an exponentially decaying kernel. The choice of kernel does not substantially alter the results presented here; therefore, we opted for the latter, as it prevents the introduction of current spiking information into the past. Additionally, the decay time was set to 5 ms to approximate the decay time of AMPA receptors, ensuring that the resulting output resembles the excitatory synaptic currents observed in downstream network layers.

## Mutual information

Mutual Information (MI) captures the dependence or shared information between two variables, providing insight into how much knowing one variable reduces uncertainty about the other. This metric is non-negative, with a value of zero indicating complete independence between the distributions of the two variables. The MI is calculated using the following expression:

$$MI = \int_X \int_Y p_{X,Y}(x,y) \log \frac{p_{X,Y}(x,y)}{p_X(x)p_Y(y)} dxdy, \tag{3}$$

where X(t) and Y(t) are continuous random variables, with probability distributions $p_X$ and $p_Y$ and joint probability distribution $p_{X,Y}$. Since the exact probability distributions are unknown and we only have a sample of them, estimating the distributions of Eq. 3 is not straightforward. To that matter, we utilized the "mutual_info_regression" function from scikit [46] which employs a nearest-neighbor approach for the estimation of the probability density's function. In cases where we calculate MI between the input and output of networks the analysis was repeated while temporally shifting the output to account for delays in the integration.

## Results

### Cross-frequency coupling and directionality depend on the connectivity

As an illustrative example and foundation for the present modeling work, we present electrophysiological recordings from the hippocampus of rats freely exploring known (familiar) and novel environments. These data revealed strong CFC between the phase of theta oscillations and the amplitude of gamma oscillations in pathway-specific local field potentials (LFPs) [12]. Fig 1a depicts the pathway-specific LFP corresponding to the CA1 *lacunosum moleculare* response to the input originating from the entorhinal cortex layer III (ECIII). The same study reported that both positive and negative CFD values coexist (Fig 1a-iii); however, the predominant directionality, $CFD_{avg}$, calculated as the average CFD across frequencies with significant CFC, was negative. This suggests that, on average, the amplitude of gamma oscillations precedes the phase of theta. Following previous studies, all CFD values reported here were renormalized to emphasize regions of high CFC, with statistical significance ($p < 0.01$) determined through surrogate analysis of clusters with the highest absolute CFD values (for more detail see the analysis section and refs. [12,29].

To investigate theta-gamma interactions, we adapted the well-established circuit motifs Interneuron Network Gamma (ING) and Pyramidal-Interneuron Network Gamma (PING) [33,35–37], with the addition of an external theta input (Fig 1b-i and 1c-i). We began by analyzing separately the two circuit motifs, named as θ-ING and θ-PING respectively. Both models included a population of pyramidal cells (PCs) excited by a θ-modulated external input, with gamma rhythms generated locally by a population of fast-spiking self-inhibitory interneurons, the basket cells (BCs), which also projected to the soma of the PCs. All connections and parameters for the two motifs were identical, except for the nature of the BC-PC inhibition: feedforward in θ-ING and feedback in θ-PING. For more details on the model see methods section.

The distinct dynamics of these two circuit motifs are shown in Fig 1b-ii and 1c-ii, where we depict the raster plots for the PCs and BCs (blue and orange dots, respectively). In the same panels we superimpose the mean somatic membrane potential ($V_{PC}$) and the mean transmembrane somatic currents $i_{transm}$, across all PCs. Both motifs exhibit strong theta-gamma CFC (see Figs 1b-ii, 1b-vi, 2b-ii, and 2b-vi) but, importantly, opposite directionalities were measured by CFD.

In the θ-ING model (see Fig 1b-v and 1b-vi), gamma activity preceded theta oscillations, resulting in a negative CFD in areas of high CFC, similar to what was observed experimentally (see Fig 1a). To investigate this further, we first ensured that the observed negative CFD was not an artifact of higher theta harmonics or non-sinusoidal waveforms, which could spuriously generate CFC and CFD values [47]. We systematically varied the transmission delay $\tau_{\theta,PC} = 30/20/10$ ms between the theta input and the PCs while keeping the transmission delay between the theta input and the BCs constant at $\tau_{\theta,BC} = 20$ ms. Thus, we effectively varied the relative transmission delay $\Delta\tau = \tau_{\theta,BC} - \tau_{\theta,PC}$ that controls the temporal relationship between gamma and theta (Fig 2). For $\Delta\tau = -10$ ms, where the theta input reaches BCs earlier than the PC dendrites, gamma bursts occur at the rise of the theta oscillation, resulting in a more negative CFD (Fig 2b, top panel). Conversely, for $\Delta\tau = 10$ ms (Fig 2b, bottom panel), gamma is relayed to PCs later, causing CFD to switch to positive. For comparison and completeness, we also considered $\Delta\tau = 0$ ms, in which theta input reaches both populations simultaneously, yet still

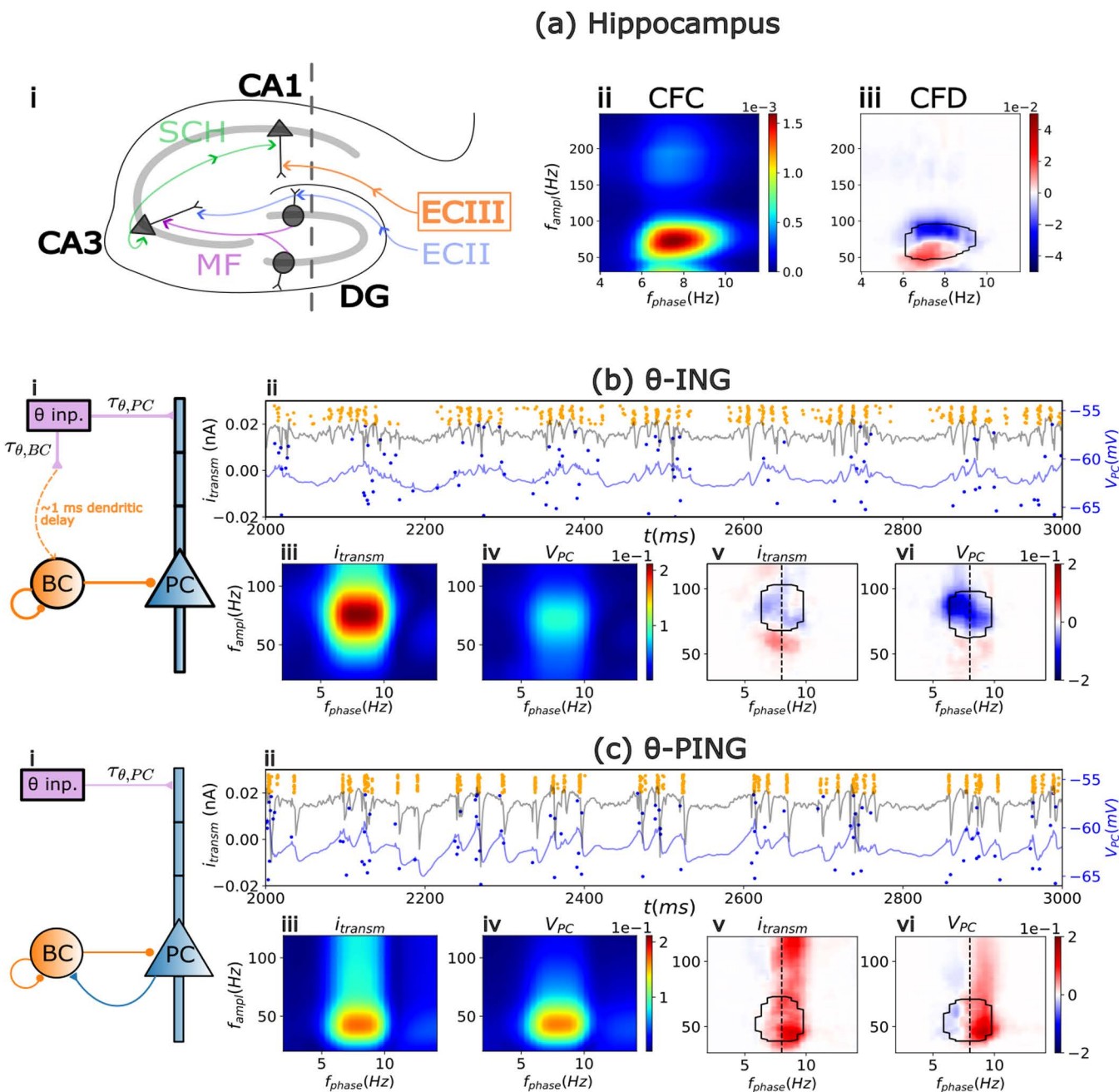

**Fig 1. Experimental and computational cross frequency coupling and directionality.** (a) Cross-Frequency Coupling (CFC) and Cross-Frequency Directionality (CFD) obtained from electrophysiological recordings from the rat hippocampus (data from [12]). (i) Schematic representation of the hippocampus with the pathway-specific layers illustrated. Sch: Schaffer Collateral, MF: Mossy Fibers, ECII and ECIII: Entorhinal Cortex layers II and III, DG: Dentate Gyrus, CA1 and CA3: Cornu Ammonis. (a-ii) Theta-gamma CFC in the field potential associated with the ECIII and (a-iii) its corresponding CFD. Note that averaging the CFD over areas of high CFC (black contour in CFD panels) shows a predominantly negative directionality $CFD_{avg}$ = -0.014. (b) θ-ING model. (i) Circuit motif. (ii) Timeseries of the average PC somatic transmembrane current ($i_{transm}$), average somatic PC membrane voltage ($V_{PC}$) and raster plots of the BCs (orange dots) and PCs (blue dots) firing. The corresponding CFC and CFD for the $i_{transm}$ and $V_{PC}$ are depicted in panels (iii and v) and (iv and vi), respectively. The black contour in CFD plots shows the 90 percentile of the respective CFC. (c) Same as (b) but for the θ-PING motif.

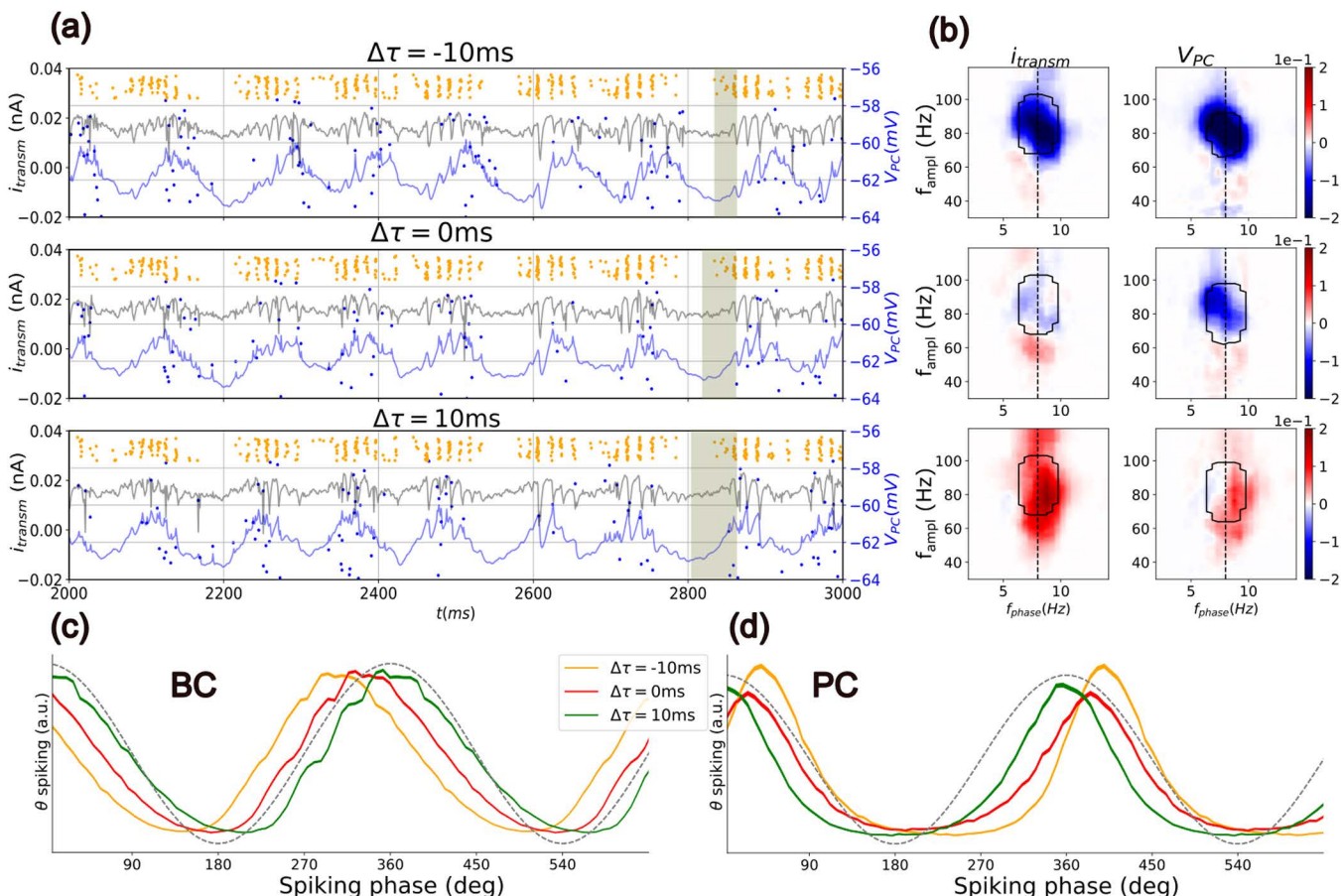

**Fig 2. Directionality in the θ-ING model reverses depending on the relative transmission delay.** (a) Timeseries of $i_{transm}$ (black line) and $V_{PC}$ (blue line) and raster plots of the BCs (orange dots) and PCs (blue dots) spikes, for different relative transmission delay Δτ. The shaded area highlights the relative displacement of the theta trough (minima of $i_{transm}$) with respect to the initiation of the gamma oscillation. (b) CFDs for $i_{transm}$ and $V_{PC}$. Contour lines indicate the range of higher CFC (more than the 90th percentile). Relative transmission delay Δτ =-10/0/10 ms increases from top to bottom in (a) and (b). In panels (c-d) we plot the theta-phase spiking distributions for BCs and PCs, respectively. For this purpose, the synaptic current generated by the θ input (indicated by the dashed black line) was used as a reference signal to extract a theta phase. The spiking profiles of the BC and PC were then calculated by assigning the corresponding phase to each spike.

results in a negative CFD (Fig 2b, middle panel; same as in Fig 1b). Overall, these results rule out a spurious contribution of theta harmonics or wave shape to the sign of CFD in the θ-ING motif.

To explain why gamma-to-theta interaction is detected when no relative transmission delay between them exists, we examined the relationship between the externally imposed theta input and the neuronal activity of both PC and BC populations (Fig 2c and 2d red curve). To make this relation clearer, we represented PCs and BCs spikes firing relative to the phase of the theta input measured in the distal dendritic compartment of the PC (see S3 Fig for the same analysis using the external θ input as reference). When Δτ = 0 ms, BC activity increases faster than the external input to the PCs (Fig 2c). Consequently, feedforward gamma inhibition reaches the PC soma before the excitatory theta activity builds up. For completeness, we also include in Fig 2c and 2d the cases for positive (in green) and negative (in orange) delays.

In contrast, in the θ-PING model, each cycle began with PC depolarization driven by the theta input, which, through the feedback connection, recruited BCs and induced gamma rhythmicity. Not surprisingly, in this configuration theta activity precedes gamma oscillations and thus a positive CFD is observed (Fig 1c-v and 1c-vi).

## Synaptic weights influence gamma-to-theta interactions in the θ-ING motif

Next, we further explored the θ-ING results [12]. To elucidate the synaptic mechanisms underlying gamma-to-theta interactions, we systematically increased the synaptic weights within the θ-ING model and evaluated their impact on CFC and neuronal theta firing patterns (Fig 3). Notably, our findings indicate that CFD in the θ-ING model remains negative across a broad range of synaptic weights (see S3 Table) and PC activity (see firing rates in S4 Table), with positive or negligible CFD values observed only at extreme parameter values.

We first examined the influence of synapses projecting to PCs. When inhibition was very high relative to excitation (Fig 3a-iv and 3b-i) PCs spiked only when the BCs' activity was at its lowest value. Thus, pyramidal spiking was scarce and out of phase with the external input, and the CFD was positive. Conversely, when excitatory input was excessively high relative to inhibition, PCs dominated the network dynamics, overriding BC-driven gamma oscillations (60–80 Hz), thereby disrupting CFC within this frequency band (Fig 3a-i and 3b-iv). This effect was particularly pronounced when synaptic excitation surpassed the dendritic spiking threshold (Fig 3b-iv). Under these conditions, the external theta generator retained the control over the network dynamics, with CFC shifting toward lower frequencies. In contrast, when excitation and inhibition were balanced, PC spiking occurred slightly after the peak of the synaptic input (Fig 3b-v). Then, the phase delay between input arrival and spiking was modulated by the level of the excitation/inhibition balance. Moreover, an increased inhibition broadened the phase distribution of PCs spiking and weakened the transmission of afferent theta signals to subsequent layers.

We next investigated the role of excitatory input to BCs (Fig 3c). This synaptic connection not only regulated BC activity and, consequently, the total inhibitory drive to PCs, but also modulated the gamma frequency of the network. An increased excitatory drive to BCs accelerated the network dynamics, a result consistent with previous computational studies of ING models [48].

For completion, we also examined variations in the connectivity of the θ-ING motif (S4 Fig), in which the inhibitory population projects to the same dendritic compartment, either proximal or distal, as the θ input. As shown in S4 Fig, the transmembrane currents in these compartments also exhibited negative CFD. Overall, our results demonstrate that θ-ING networks exhibit flexible dynamics, capable of modulating both single-cell and population-level activity by changing synaptic weights (synaptic plasticity), while remaining in a mode where gamma leads theta locally.

## Circuits of combined θ-ING and θ-PING motifs

Given that brain circuits are generally endowed with both feedforward and feedback inhibition simultaneously, we next investigated the predominant theta-gamma directionality in the combined model, by adding the feedback PC→BC connection to the θ-ING model or, equivalently, the feedforward θ→BC connection to the θ-PING model. Under these conditions, we found that theta-gamma directionality was primarily determined by feedforward inhibition, as indicated by changes in CFD when the strength of the θ→BC connection is varied (Fig 4a). In contrast, changing the strength of the feedback PC→BC connection resulted in significantly less pronounced changes (Fig 4a).

To understand better the differential role of these connections, we focused on the dynamics of motifs transitioning from a pure θ-PING model to a mixed model with stronger feedforward connections (Fig 4b). By increasing the external θ excitatory drive to BCs, this input overcomes the excitation received from PCs through the feedback PC→BC connection. Consequently, BCs fire progressively earlier relative to the theta phase, resulting in a predominant gamma-to-theta directionality (see blue rectangle in Fig 4b). Simultaneously, the earlier inhibition of PCs further diminishes the influence of the feedback connection (PC→BC). Therefore, a unified model incorporating both feedforward and feedback connections suggests that CFD can be flexibly controlled via a local feedforward inhibitory connection.

## Evidence of θ-ING/θ-PING motifs in experimental data

The results obtained from the model provided at least two testable predictions. First, there is a relationship between CFD and the frequency of gamma oscillations coupled with theta, such that the more negative the CFD, the higher the gamma

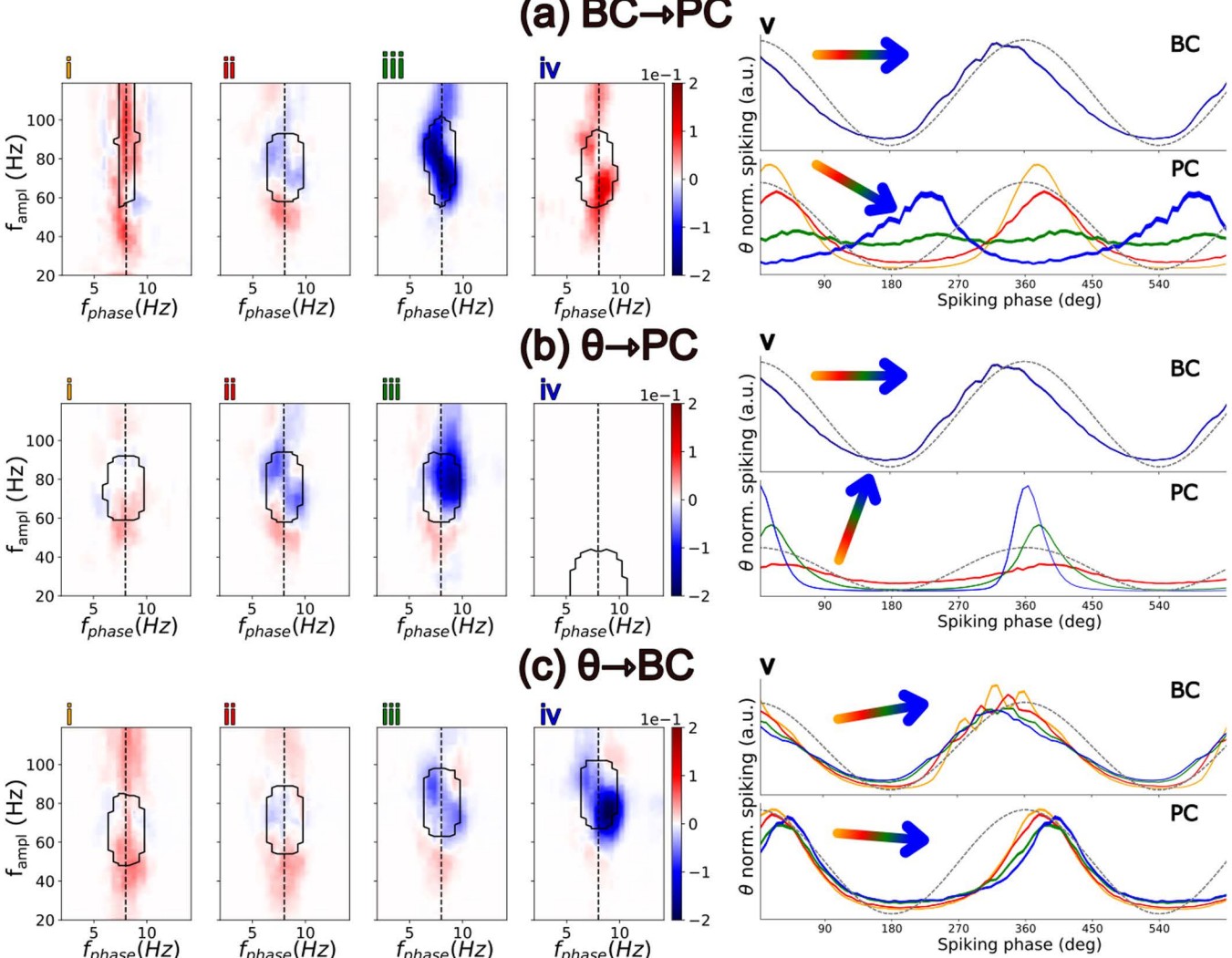

**Fig 3. Role of different synaptic coupling strengths on directionality, firing rates, and spiking phase.** The synaptic coupling takes four increasing values: $w_i < w_{ii} < w_{iii} < w_{iv}$ (see S3 Table for exact values) for the following connections: BC→PC (a), θ→PC (b), and θ→BC (c). (i-iv) CFD with the contour encircling areas of high CFC values. (v) Spiking distribution as a function of the theta phase for BCs (Top) and PCs (Bottom). Spikes are binned according to the phase of the synaptic current of the θ input in the PCs dendrites (same as in Fig 2). For visualisation purposes, each spiking distribution was normalized so that the integral over a theta cycle is 1. The orientation angle of the coloured arrow indicates whether the firing rate increases, decreases, or remains the same as the corresponding synaptic weight increases (see S4 Table for exact values).

frequency (Fig 5a). This relationship arises because in θ-PING, unlike in θ-ING, gamma generation depends on the spiking of PCs which are intrinsically slower than BCs. As the strength of the feedforward input to BCs increases, gamma generation becomes less dependent on PC firing, while its frequency progressively increases (Fig 4a). Second, the quantitative CFD value depends on the balance between feedforward and feedback inhibitory pathways, making it potentially sensitive to different cognitive states.

Next, we sought to test these predictions using previously acquired electrophysiological data recorded from the hippocampus of 5 rats freely exploring familiar and novel spatial arenas (see Methods for details). We reasoned that exploring these distinct environments, which impose different cognitive demands—retrieving previously acquired spatial information

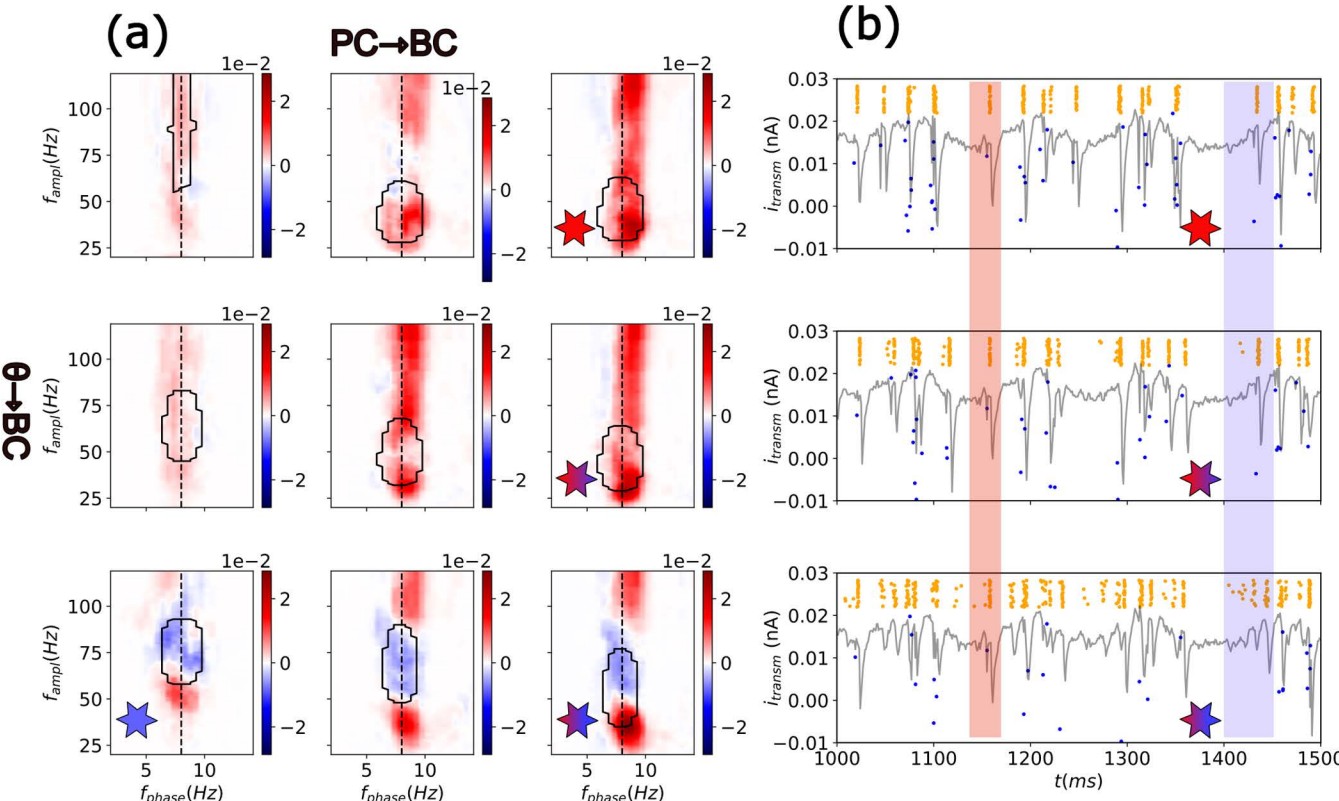

**Fig 4. Transitions between positive and negative CFD in a combined θ-ING and θ-PING model.** (a) CFD in the 2D parameter space of the relevant synaptic weights PC→BC and θ→BC (see S5 Table). The θ→BC input strength increases in rows from top to bottom. The PC→BC input strength increases in columns from left to right. The red and blue stars denote the parameters of the pure θ-PING and θ-ING motifs, respectively (same motifs shown in Fig 1). Stars of mixed colors depict the transition from a pure θ-PING to mixed models by increasing the feedforward connection strength θ→BC. (b) Transmembrane currents and raster plots of the θ→BC transition (same stars code as in panel a). Coloured rectangles highlight the initial part of a θ oscillation as well as the first few gamma cycles. The rectangle in blue depicts an example of directionality transition from top (CFD > 0) to bottom (CFD < 0). Note how increasing feedforward inhibition advances the firing of BCs (orange dots in the raster plot), and consequently the gamma oscillations over the theta cycle and the firing of PCs (blue dots). The rectangle in red depicts one example of theta cycle in which transition was not fully realized, highlighting the dynamic character of the CFD.

versus encoding new information in a novel environment—would provide an effective testbed for identifying dynamic regulations of CFD. Indeed, this dataset revealed statistically significant differences in theta-gamma CFC [12].

We tested our predictions in the experimental dataset and found that, indeed, CFD was significantly modulated by the behavioural task, showing more negative CFD values during the exploration of the novel environment compared to the familiar one (Fig 5b and 5c). In parallel, the gamma frequency exhibited a tendency to increase across all animals during the novelty condition (Fig 5d), although this effect did not reach statistical significance (p = 0.140). However, we found a significant association between CFD and gamma frequency across all animals, consistent with model predictions, showing that more negative CFD values were linked to higher gamma frequencies in the hippocampus (Fig 5e).

## Functional differences of θ-ING and θ-PING motifs

Given the experimental evidence of the dynamic behaviour of cross frequency interactions, we explored whether different directionality modes, ranging from pure θ-ING to pure θ-PING, with two intermediate mixed motifs (Cases 2 and 3, Fig 6a), exhibit distinct computational properties. Recognizing that in natural circuits, more than one pathway may converge

PLOS Computational Biology

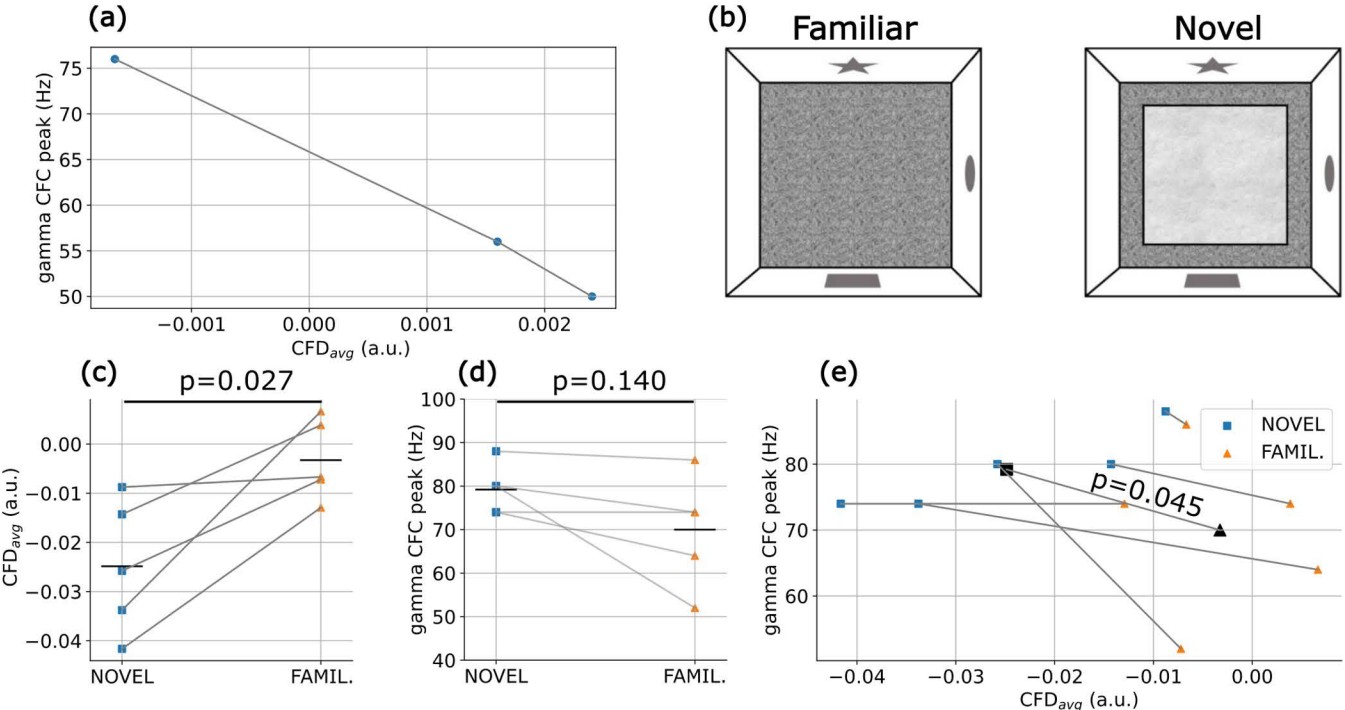

**Fig 5. Experimental test of model predictions.** (a) Mixed θ-ING/θ-PING motifs show that more negative CFD values are associated with CFC peaking at higher gamma frequencies (same motifs as in the middle panel of Fig 4a). (b) Schematic of the experimental setup: after one week of habituation with daily sessions in an open field, animals (n = 5) were introduced to a novel floor surface (mismatched novelty environment). Hippocampal recordings were performed during free exploration of both environments [12]. (c) CFD values were significantly lower during exploration of the novel environment compared to the familiar one (paired t-test t(4) = -3.40, p = 0.027). (d) During exposure to novelty, CFC tended to peak at higher gamma frequencies compared to the familiar condition, although this effect did not reach statistical significance (paired t-test t(4) = 1.84, p = 0.140). (e) Combined data from (c) and (d) represented in two dimensions. The slopes of all pairs of observations per animal (connected colored symbols) are significantly negative (one-tail t-test t(4) = -2.22, p = 0.045), demonstrating the association between CFD values and peak gamma frequencies. Black symbols represent average values across all rats for both conditions.

into a neural population, we extended our analysis to include two external input pathways: (1) a θ-driven input targeting the distal dendrites of the PCs, as in previous analyses, and (2) a Poisson-distributed spiking input, termed the parallel pathway, targeting the proximal dendritic segment (see methods for more details).

To evaluate the computational properties of each motif, we employed two protocols. The first one assessed the signal fidelity by calculating the mutual information (MI) between an external input and its corresponding PC firing output. The second protocol evaluated the robustness of an input to variability. This is measured as the MI between two outputs generated in response to the same input, either the θ or the parallel pathway, while all other inputs varied (see Fig 6b and the methods section for details on the protocols).

While all four motifs' output effectively shared information from both the θ and parallel pathway inputs, they did so in a complementary manner. PING-shifted configurations exhibited higher signal fidelity with the θ input, the primary driver of the circuit (Fig 6c), whereas ING-shifted configurations exhibited higher fidelity with the parallel pathway (Fig 6d). This disparity between motifs arises from a fundamental difference, evident in the membrane potential traces shown in Fig 1b and 1c. In the pure θ-PING case, PCs and BCs are strongly coupled via feedback connections, resulting in larger gamma oscillations than in the θ-ING motif. This creates a robust theta-gamma scaffold where opportunity windows for information transmission are restricted to the peaks of the gamma cycles nested within each theta cycle. Inputs that are

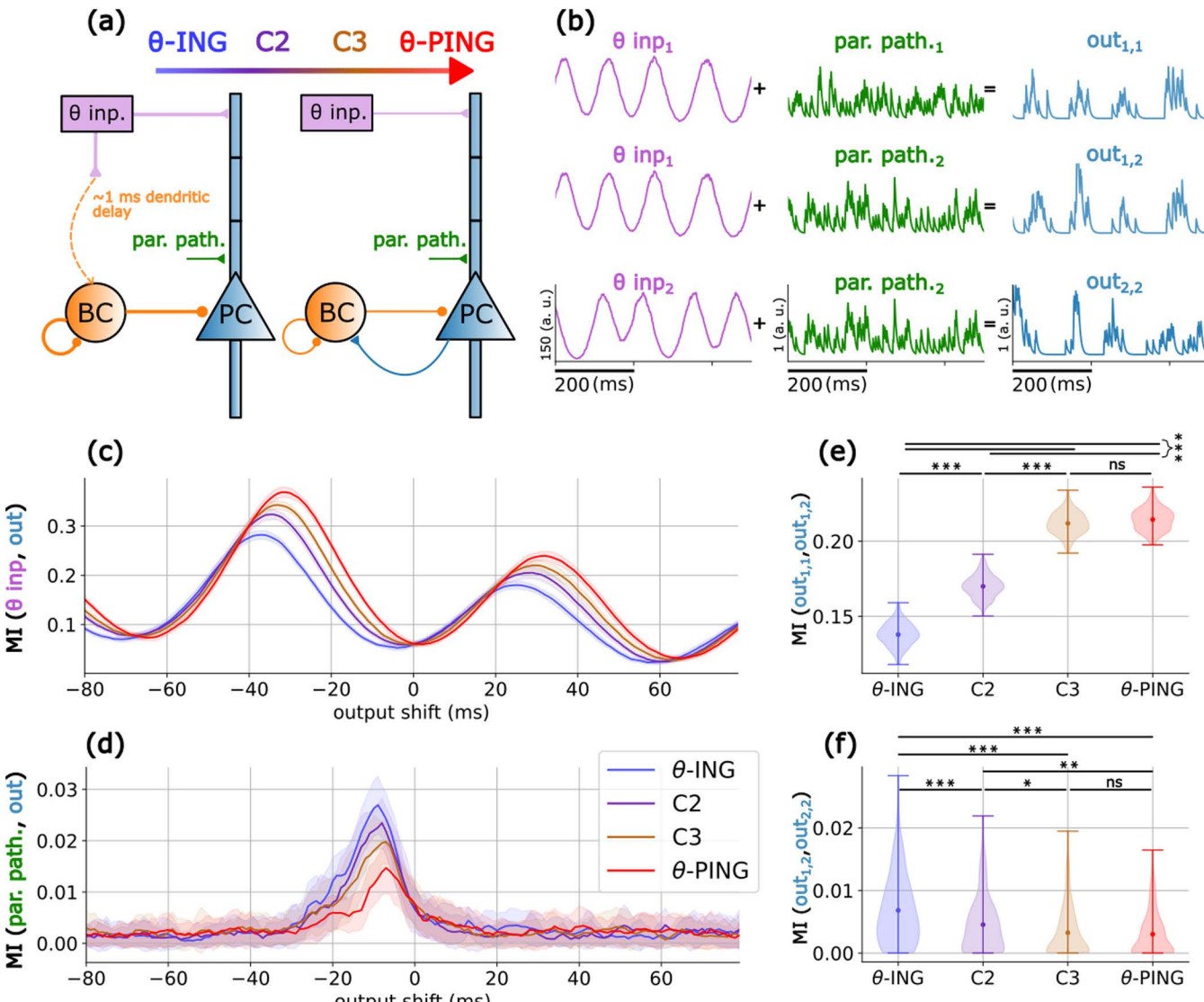

**Fig 6. Functional analysis of θ-ING and θ-PING motifs.** (a) θ-ING and θ-PING motifs with the parallel pathway introduced at the proximal dendritic segment of the PCs. Cases 2 and 3 (C2 and C3, respectively) are motifs in which the PC→BC and θ→BC synaptic weights fall between the values associated with the θ-ING and θ-PING motifs (see S6 Table). (b) Schematic representation of the inputs in the θ (purple) and parallel (green) pathways and the outputs produced by the PCs (blue). All time series were generated by convolving the spike trains with a 5 ms exponential decay kernel. (c) Mutual Information (MI) between the θ input and the pyramidal output. Pairwise t-tests with Bonferroni correction for multiple comparisons between the maximum MI values showed that all cases are significantly different (p < 0.001). (d) Same as in (c) but considering the parallel pathway as input. All cases are significantly different (p < 0.001) except for C3 vs. θ-PING (p = 0.0104), θ-ING vs. C2 (p = 0.273) and C2 vs.C3 (p = 0.103). (e) Consistency measured as MI between pairs of outputs in response to the same θ input while all other inputs changed. (f) Same as in (e) but considering the parallel pathway. Stars in panels (e) and (f) depict significant differences using pairwise t-tests with Bonferroni correction: ***/**/* p < 0.001,0.01,0.05 respectively.

not synchronized with this rhythmic theta-gamma structure, such as the parallel pathway, more often fail to evoke PC responses. In contrast, in the θ-ING motif, gamma oscillations are primarily driven by BCs in a feedforward manner and, while still nested within the theta cycle, this gamma activity is more loosely anchored to PC firing, broadening the temporal windows for information transmission. Consequently, strong perturbations falling within or outside the theta-gamma coding

framework are able to overcome depolarization thresholds. To support this interpretation, we conducted simulations where the parallel pathway was replaced with single pulse perturbations. This allowed us to assign a gamma phase to each perturbation, revealing that the θ-PING motif was responsive during the peak of the gamma oscillation, whereas the θ-ING motif exhibited firing throughout the cycle (see S5 Fig), confirming our hypothesis.

Finally, we assessed the second protocol, namely the robustness of an input to variability. Higher fidelity of the θ-input in PING-shifted motifs (Fig 6c) was accompanied by greater consistency (Fig 6e). Similarly, improved fidelity of the parallel pathway in ING-shifted motifs (Fig 6d) was also more robust to variability (Fig 6f). Overall, the continuous transition observed between the two models, modulated by the balance between feedforward and feedback inhibition, suggests a mechanism for weighting or selecting communication channels with different temporal structures, which could be reflected in the CFD index.

## Discussion

In this study, we have shown that the directionality of local theta-gamma interactions depends on, and can be regulated by, the feedforward and feedback inhibition balance. Specifically, we found that motifs with enhanced theta-modulated feedforward inhibition (θ-ING) exhibit dominant gamma-to-theta interactions, while those with enhanced theta-modulated feedback inhibition (θ-PING) display dominant theta-to-gamma interactions. In a combined circuit containing both feedforward and feedback connections, we found that the feedforward connection determines theta-gamma directionality and governs the transition between motifs or operational modes.

We further found that these operational modes significantly influence the firing phase of PCs within the theta cycle. The fine-tuning of gamma-theta interactions defines the temporal windows within the theta cycle in which cell assemblies can be formed. In this manner, the feedback-feedforward inhibitory balance may implement a push-pull mechanism that governs the firing phase of PCs: increased feedforward inhibition leads to phase precession and increased feedback inhibition induces phase recession. For the same reason, different operational modes favor the response of PCs to inputs with different temporal structures. By fine-tuning the timing between slow and fast oscillations, this mechanism selectively prioritizes information transmitted through different afferent pathways, enhancing computational flexibility.

An additional notable observation was that the timing of theta-nested gamma inhibition modulated the phase difference between the afferent theta rhythm and the local theta spiking profile (Fig 2d). This mechanism may facilitate the coordination of theta rhythms across pathway-specific field potentials, as observed experimentally [12]. Importantly, our analysis of hippocampal electrophysiological data supports this dynamical framework, demonstrating a behavioral state-dependent regulation of cross-frequency interaction directionality. In the following sections, we explore the practical applications and limitations of our model and discuss its relevance to experimentally observed brain dynamics.

### Is θ-ING a realistic model for negative CFD?

The emergence of negative cross-frequency directionality in the θ-ING motif is critically dependent on the rapid response of BCs, enabling gamma oscillations to precede the locally measured theta rhythm. This raises an important question regarding the physiological validity of the conditions assumed in our model. We argue that our implementation of BC activation times is, in fact, conservative.

First, dendritic transmission in BCs is notably rapid, with evoked postsynaptic currents at distal dendritic sites depolarising the soma within less than 1 ms dendritic delay [41]. Second, stimulation of the Schaffer collateral induces monosynaptic excitation followed by disynaptic inhibition after only 1.9 ± 0.2 ms delay [49]. This is faster, on average, than the delays assumed in our model, which consists of the 1 ms dendritic delay plus 1.5 ± 0.2 ms of synaptic delay from BC to PCs and the spike generation time. Third, both *in vivo* and *in vitro* patch-clamp recordings from the CA1 region have demonstrated faster action potential initiation in BCs compared to PCs following stimulation of pathways that simultaneously excite BCs and PCs, such as the alvear pathway and the Schaffer collateral, respectively [50,51]. Finally, regarding population

dynamics, *in vivo* recordings from mice running in a maze have shown that interneurons activity in the pyramidal layer of CA1 peaks approximately 60 degrees (or equivalently 20 ms) ahead of the theta phase of PCs (see Fig 5 in ref. [52]). This finding is consistent with our results (Fig 2c and 2d), which show that BCs lead PCs significantly in theta phase. Taken together, these findings support the plausibility of the inhibitory time delays implemented in our model.

## Experimental evidence and interpretation of CFD measurements

Our results indicate that the balance between feedforward and feedback inhibition in local circuits determines the directionality of cross-frequency interactions. We further showed that a feedback-shifted balance favors transmission in an afferent pathway by promoting the specific cross-frequency rhythmicity driven by the afferent input, while a feedforward-shifted motif broadens the opportunity window for encoding, facilitating parallel pathways to transmit. Accordingly, dynamic CFD measures could be interpreted in terms of predominant inhibitory circuit motifs and flexible prioritization of functional connectivity pathways.

In the rat hippocampus, we have found that during novelty exploration, activity in the EC→CA1 connection exhibits more negative CFD values than during exploration of familiar environments. According to our model, a less negative CFD corresponds to a PING-shifted state that favors transmission along the main afferent pathway (here the EC→CA1 connection) while decreasing competition from parallel pathways, such as the Schaffer collateral (driven by the trisynaptic DG→CA3→CA1 connections). In contrast, a more negative CFD suggests an ING-shifted state, where parallel inputs gain greater relevance. This interpretation aligns with previous findings from c-Fos interaction networks, which reported an activity shift from a dominant EC→CA1 monosynaptic pathway during familiar conditions to a dominant trisynaptic pathway during novelty exploration [53]. Animals in the mismatch novelty condition compare the contextual representation retrieved from memory with incoming sensory information. Consistent with established perspectives on information processing within the hippocampal circuit [12,22,54,55], our model results indicate that novelty conditions favor an ING-shifted operational mode. This mode facilitates the integration of parallel pathways, enabling the retrieval of memories (CA3→CA1) while simultaneously processing environmental cues and encoding new information (EC3→CA1).

Furthermore, a recent study using electrocorticography in human epilepsy patients performing a spatial attention task reported a relationship between alpha-gamma CFD values and attentional states [30]. Specifically, more negative CFD values were associated with non-attended stimulus, while lower absolute (but still negative) CFD values were linked to attended stimulus. According to our model, less negative CFD would be reflecting a PING-shifted circuit state that favors transmission of the message in the afferent (attended) pathway, in this case, the visual dorsal stream, reducing the impact of parallel (distracting) inputs. Similarly, a more negative CFD, as found for non-attended stimulus, is expected from an ING-shifted circuit in which parallel inputs gain relevance over the afferent pathway. Importantly, the authors reported enhanced functional connectivity along the visual dorsal stream in the first case, and suppressed connectivity in the second, supporting our interpretation. The model offers a mechanism to explain attention deployment dynamically and flexibly, based on feedforward-feedback inhibitory balance and reflected in the CFD metric.

Interestingly, these studies also confirmed key mechanistic predictions of our theoretical framework. First, in the electrocorticography study alpha-phase activity in upstream regions preceded downstream high gamma activity, while locally, gamma preceded alpha, which aligns well with our model predictions (compare S3 Fig and 2 in our manuscript and Fig 2 of [30]). Second, both studies provide evidence supporting the mechanism of directionality control based on the feedforward and feedback inhibitory balance, showing that more negative CFD values are associated with higher peak gamma frequencies, as expected from the feedforward recruitment of BCs (Fig 5 in our manuscript and Fig 2 in [30]).

## Limitations

Our model is intentionally minimal, designed to ensure the reported directionalities are broadly applicable across different brain regions, as supported by existing literature [12,29–31,56–58]. This generalizability, however, comes at the expense

of region-specific precision. While addressing such specificity would require detailed multicompartmental models tailored to particular brain areas [59–61], the simplicity of our model allows it to capture fundamental dynamics effectively. Future work incorporating diverse interneuron types and their unique connectivity patterns would further refine our understanding [62].

One region where achieving greater specificity may be particularly important is the hippocampus, given the ongoing debates surrounding the mechanisms underlying the origin of CFC. As discussed in the Introduction, competing hypotheses propose that CFC may either be generated locally within the hippocampus or inherited from upstream regions projecting onto distinct dendritic compartments. Interestingly, our model supports both mechanisms simultaneously, suggesting cooperation rather than competition between global and local processes. In the model, CFC can emerge from the interaction between an external theta source and locally generated gamma oscillations at the somatic layer or within distinct dendritic compartments (see S4 Fig), consistent with the hypotheses proposed in [24,26,63]. Furthermore, the output of PCs in the same model exhibits cross-frequency coupled activity that can be relayed to downstream layers (see S2 Fig), aligning with the perspectives presented in [22,23,64]. Our framework also captures a graded modulation of perisomatically generated gamma rhythms, in line with the flexible gamma activity described by [8]. This adaptability highlights the model's utility as a testbed for future investigations seeking to disentangle local versus network contributions to CFC and to understand how these mechanisms influence firing sequences and information routing.

While the motifs reproduce gamma-to-theta and theta-to-gamma directionalities, they do not account for large cross-frequency lags, such as the -50 ms lag between alpha and beta rhythms observed in human auditory cortex electrocorticogram data [31]. However, by accounting for plausible relative transmission delays, as illustrated in Fig 2, the motifs can accommodate a wide range of experimentally observed cross-frequency lags. These lags may reflect multi-synaptic pathways. For instance, in the CA3 region, long-lag negative CFD could arise if BCs are monosynaptically recruited by entorhinal cortex inputs, while PCs are driven through the disynaptic circuit (EC→DG→CA3). Investigating these delays in specific anatomical contexts offers promising opportunities for future research.

In summary, despite its simplifications, the model offers a robust framework for understanding CFC and CFD across regions while paving the way for more detailed, region-specific explorations in future studies.

## Concluding remarks

In conclusion, our combined modeling and experimental results propose a mechanism for the flexible gamma-to-theta interactions observed in electrophysiological recordings. They support the view that both θ-ING/θ-PING operational modes exist along a continuum rather than as mutually exclusive alternatives and suggest a functional role for feedforward/feedback inhibitory balance in prioritizing parallel information pathways converging onto the same dendritic tree. Thus, CFD and related measures may serve as valuable experimental entry points for further elucidating these processes.

## Supporting information

**S1 Table.** Synaptic weights for Fig 1. The synaptic weight depicted for PC→BC is for the θ-PING, otherwise it is zero. Similarly, the synaptic weight depicted for θ→BC is for the θ-ING, otherwise it is zero. The difference in the order of magnitude between PC→BC and θ→BC is due to differences in the number of presynaptic neurons (80 vs 500) and their activity (0.49Hz vs 8Hz).
(PDF)

**S2 Table. Synaptic parameters of the model.** NMDA has an additional scaling factor due to the magnesium block $1/(1 + 0.28[Mg]e^{-0.062V})$, where [Mg] = 1mM is the concentration of magnesium, and V the membrane potential in mV.
(PDF)

**S3 Table. Synaptic weights for Fig 3.** All other synaptic parameters are as in θ-ING (see S1 Table).
(PDF)

**S4 Table. Mean firing rates for BCs ($f_{r,BC}$(Hz)) and PCs ($f_{r,PC}$(Hz)) in the θ-ING model for different synaptic weights $w_i < w_{ii} < w_{iii} < w_{iv}$ in the circuit connection (Conn.).** The simulations are the same as in Fig 3. Blue-colored rates are associated with motifs that exhibit negative CFD.
(PDF)

**S5 Table. Synaptic weights for Fig 4.** As also stated in S1 Table, the difference in the order of magnitude between PC→BC and θ→BC is due to differences in the number of presynaptic neurons (80 vs 500) and their activity (0.49Hz vs 8Hz).
(PDF)

**S6 Table. Synaptic weights for Fig 6.**
(PDF)

**S1 Fig. Directionality and coupling in synthetic data.** (a) Synthetic data of different CFD made following [29]. The θ signal was generated by concatenating sinusoidal segments of mean amplitude 10 and different periods drawn from a Gaussian of mean 125ms. The γ signal is an 80Hz sinuisoidal coupled to the trough of θ. Then, θ and γ are added together to create 3 timeseries with different theta-gamma relationship: in the blue line γ is lagged 10ms behind θ, in the green timeseries γ is not lagged whereas for the red line θ laggs behind γ by 10ms. (b) Top: Cross Frequency Coupling of the timeseries in (a). Notice that independently of the time difference between theta and gamma all timeseries have identical CFC peaking at (8Hz,80Hz). (c) Bottom: Cross Frequency Directionality of the lines in (a). CFD is able to detect the time delay between the γ amplitude and θ phase.
(TIFF)

**S2 Fig. Generation of the PC output of the network.** (i) Raster plots of PCs in blue (and BCs in orange for completeness). The convolution of each PC spike with either a 1ms Gaussian kernel (ii) or a 5ms decaying exponential kernel (iii) generates a proxy for the instantaneous output firing rate (in arbitrary units). A CFC analysis of the firing rate (iv, v) provides insight into what could be detected in the dendrites of a downstream layer which is not dependent on the kernel. This analysis was conducted for PC spikes in a θ-ING motif (a) and a θ-PING motif (b). To ensure sufficient spikes for the CFC analysis, both networks were scaled up by a factor of 4—achieved by increasing the number of neurons and projections while reducing the number of synaptic weights.
(TIFF)

**S3 Fig. Same analysis as in Fig 2, but using the external θ drive phase as the θ reference for spiking and CFD calculations.** Panels (a) and (b) show the θ phase of PC and BC spiking, respectively. Panel (c) illustrates the CFD for $i_{transm}$ (top) and $V_{PC}$ (bottom). To derive the θ phase of the external population, spikes are passed through a decaying exponential kernel with a 5ms time constant. Note that when using the external population's θ phase, the BC phase remains unchanged, while the PC phase shifts significantly due to different offsets. Finally, as the local γ is consistently generated by the external θ driver, the CFD remains positive under all conditions.
(TIFF)

**S4 Fig. Different θ-ING connectivities retain negative CFD.** (a) The θ-ING motif analyzed in this study is visualized for reference (same as in Fig 1b-v) with the additional CFD of the transmembrane currents at the proximal and distal dendrites. (b) The theta input excites the BC and PC population closer to their somata. (c) The inhibitory population is positioned and projects at the same layer as the theta input.
(TIFF)

**S5 Fig.** Single perturbation MI analysis. A single spike is introduced in the proximal dendrite at a predefined time within the interval (1-1.5) s (here at 1 s, indicated by the black dashed line). Panels (a) and (b) show the dynamics for a θ-ING

and a θ-PING motif, respectively, both with similar firing rates. Blue lines represent the mean membrane potential at the PC soma ($V_{PC,g}$) for the unperturbed case, while gray lines show the evolution after the perturbation ($V_{PC,p}$). Open circles indicate spikes in the perturbed simulations (gray for PCs, brown for BCs), and solid circles represent spikes in the baseline condition (blue for PCs, red for BCs). Since only one perturbation is applied, the resulting encoding value between output and perturbation can be related to the network state at the time of perturbation. (c) Encoding values are plotted against the θ phase of $V_{PC,p}$, with their histogram overlaid, where 180° represents the trough and 0°/360° the peak. (d) Same as (c), using a high-pass filter cutting of frequencies lower than 20 Hz to capture the gamma activity of both motifs. (e) Same as (d), but for encoding values only when the θ phase is between -90° and 90°, i.e., when the network is more depolarized by the θ input. In both panels (e) and (d) θ-PING MI depends on the γ phase more than θ-ING. (TIFF)

## Acknowledgments

We thank Spyridon Chavlis, Panayiota Poirazi and Oscar Herreras for their valuable comments.

## Author contributions

**Conceptualization:** Víctor J. López-Madrona, Santiago Canals, Claudio R. Mirasso.

**Data curation:** Dimitrios Chalkiadakis, Víctor J. López-Madrona.

**Formal analysis:** Dimitrios Chalkiadakis, Jaime Sánchez-Claros, Víctor J. López-Madrona.

**Funding acquisition:** Santiago Canals, Claudio R. Mirasso.

**Investigation:** Dimitrios Chalkiadakis, Jaime Sánchez-Claros, Víctor J. López-Madrona, Santiago Canals, Claudio R. Mirasso.

**Methodology:** Dimitrios Chalkiadakis, Jaime Sánchez-Claros, Víctor J. López-Madrona, Santiago Canals, Claudio R. Mirasso.

**Project administration:** Santiago Canals, Claudio R. Mirasso.

**Resources:** Santiago Canals.

**Software:** Dimitrios Chalkiadakis, Jaime Sánchez-Claros, Víctor J. López-Madrona.

**Supervision:** Víctor J. López-Madrona, Claudio R. Mirasso.

**Validation:** Santiago Canals.

**Visualization:** Dimitrios Chalkiadakis, Jaime Sánchez-Claros.

**Writing – original draft:** Dimitrios Chalkiadakis, Jaime Sánchez-Claros, Víctor J. López-Madrona, Santiago Canals, Claudio R. Mirasso.

**Writing – review & editing:** Claudio R. Mirasso.

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
