## [Decision Letter · Decision Letter 0]

28 May 2025

PCOMPBIOL-D-25-00474

The Role of Feedforward and Feedback Inhibition in Modulating Theta-Gamma Cross-Frequency Interactions in Neural Circuits

PLOS Computational Biology

Dear Dr. Mirasso,

Thank you for submitting your manuscript to PLOS Computational Biology. After careful consideration, we feel that it has merit but does not fully meet PLOS Computational Biology's publication criteria as it currently stands. Therefore, we invite you to submit a revised version of the manuscript that addresses the points raised during the review process.

Please submit your revised manuscript within 60 days Jul 28 2025 11:59PM. If you will need more time than this to complete your revisions, please reply to this message or contact the journal office at ploscompbiol@plos.org. Please include the following items when submitting your revised manuscript:

We look forward to receiving your revised manuscript.

Kind regards,

Roman Bauer

Academic Editor

PLOS Computational Biology

Lyle Graham

Section Editor

PLOS Computational Biology

**Additional Editor Comments :**

Based on the reviewers' comments this manuscript requires a major revision. Please make sure to address all the raised issues.

**Journal Requirements:**

4) We notice that your supplementary Figures, and Tables are included in the manuscript file. Please remove them and upload them with the file type 'Supporting Information'. Please ensure that each Supporting Information file has a legend listed in the manuscript after the references list.

1) If the funders had no role in your study, please state: "The funders had no role in study design, data collection and analysis, decision to publish, or preparation of the manuscript."

2) If any authors received a salary from any of your funders, please state which authors and which funders.

6) Please ensure that the funders and grant numbers match between the Financial Disclosure field and the Funding Information tab in your submission form. Note that the funders must be provided in the same order in both places as well. Currently, the order of the grants is different in both places. 

7) Please provide a completed 'Competing Interests' statement, including any COIs declared by your co-authors. If you have no competing interests to declare, please state "The authors have declared that no competing interests exist". 

**Reviewers' comments:**

Reviewer's Responses to Questions

Reviewer #1: Chalkiadakis et al. provide insights into the directionality of theta-gamma interactions in the hippocampus, focusing on the perspective of principal cells (PCs). They show that feedforward inhibition biases PC output toward gamma-to-theta directionality, whereas feedback inhibition induces the opposite, theta-to-gamma directionality. In a more realistic motif that integrates both ING and PING elements, they demonstrate that varying the relative strength of these components produces a continuum of gamma frequencies and theta-gamma directionalities. Their interpretation links these outputs to the interaction between theta inputs, basket cell activity, and PC responses. I find this an interesting theoretical contribution, but I have several conceptual comments regarding the manuscript, which I detail below.

In the experimental results, CFD analyses focus on the LFP component associated with EC inputs to CA1 (stratum lacunosum-moleculare). However, López-Madrona et al. (2020) also applied ICA to isolate pathway-specific components corresponding to CA3 inputs via the Schaffer collaterals (stratum radiatum). Given that CA3 is arguably the dominant excitatory drive to CA1, could the authors clarify why the EC-associated component was prioritized for the analyses shown in Figure 1? This question becomes particularly relevant given the authors’ interpretation favoring an ING motif. While there is strong anatomical and functional evidence that CA3 inputs provide direct excitatory drive to CA1 PV-positive interneurons, I am not aware of comparably strong evidence for substantial direct excitation of these cells by EC inputs.

Related to this, the authors write in the discussion (lines 530–533): “Third, both in vivo and in vitro patch-clamp recordings from the CA1 region have demonstrated faster action potential initiation in BCs compared to PCs following stimulation of pathways that simultaneously excite BCs and PCs, namely the perforant path and the Schaffer collateral, respectively [48, 49].” While I agree that these papers demonstrate earlier BC spiking in response to Schaffer collateral stimulation, they do not appear to examine responses to perforant path input. As such, mentioning the perforant path hinting EC inputs to CA1 basket cells may be misleading. Indeed, basket cell dendrites typically do not extend into stratum lacunosum-moleculare. Although the alvear pathway might in principle mediate EC excitation of these interneurons, any such mechanism would require appropriate citation or further elaboration.

In sum, if a theta-modulated input is responsible for driving ING-like gamma in CA1, the CA3 pathway appears to be a more plausible candidate than EC. I may be overlooking a key aspect, and would welcome clarification or correction. Alternatively, if the EC component was selected for technical or interpretative reasons, it would be helpful for the authors to discuss this caveat and/or report CFD results based on the CA3-associated component as well.

Regarding “Cross-frequency coupling and directionality depend on the connectivity”:

In your models, the initiating driving force of circuit activity is the theta current, which eventually triggers gamma activity. Thus, I would interpret the ground truth (global) directionality as being from theta to gamma in both models. However, the analyses in Figure 1b,c involve CFD computation using local PC signals (namely itransm and VPC), rather than a global network measure. If I understand correctly, in the ING model, basket cells respond more quickly to theta inputs and impose gamma-paced inhibitory currents on PCs, causing gamma to precede theta at the PC soma. In other words, the negative CFD observed for the ING model is exclusively from the point of view of PCs. Although the explanation based on interneuron latency and dendritic integration is plausible, it would be helpful to emphasize more clearly that CFD in this context reflects local temporal precedence rather than external causality. That being said, comparing model-derived CFD (from somatic itransm and VPC) with experimental CFD (based on LFPs) requires further justification. It is not obvious that intracellular somatic signals are equivalent to pathway-specific LFP components in terms of how CFD should be interpreted. Could the authors clarify if I am missing something, or explain why the LFP independent component can be considered analogous to the local PC signals used in the model for interpreting CFD?

The methods for the experimental analysis presented in Figure 5 are underspecified. It is unclear how gamma frequency was estimated, what signal was used (e.g., an ICA component or raw LFP), and which anatomical layer or gamma band was analyzed. Importantly, previous work from the same group (López-Madrona et al., 2020), based on the same dataset, demonstrated the presence of multiple coexisting gamma oscillations (e.g., slow, mid, and fast gamma) across distinct ICA components, presumably reflecting different afferent pathways; a view consistent with a substantial body of literature on hippocampal gamma oscillations. Based on the frequency bands analyzed here, the authors appear to focus primarily on what the discrete-gamma framework would classify as mid gamma, typically associated with EC inputs. However, it is also valid to conceptualize mid gamma as an locally generated rhythm, as appears to be the case in the present study, aligning more closely with the continuous-gamma paradigm proposed by Douchamps et al. (2024). If this is indeed the intended framework, it would be helpful for the authors to make this conceptual shift explicit and to provide some justification for adopting it rather than just commenting this quickly in Concluding remarks. Alternatively, the same analysis could be extended to other well-characterized gamma bands to assess generalizability. In any case, these distinctions should be discussed more clearly, as they are critical for interpreting the physiological relevance of the model.

The authors refer to their MI analysis as assessing the “encoding capacity” of each model configuration. This terminology is somewhat misleading. As I understand it, the MI (as it is applied here) reflects the extent to which individual pyramidal cell outputs track a specific input signal, essentially quantifying signal fidelity rather than encoding capacity. Thus, high MI in this analysis could result from a trivial scenario in which all PCs reliably and redundantly follow a global input (with a delay). Can we refer to this as high “encoding capacity”? Conventionally, encoding capacity implies the ability to represent or discriminate between multiple distinct input patterns, to flexibly engage different subsets of PCs depending on the structure of a multidimensional input, or to learn and represent specific population input patterns in the PC outputs so they can be distinguished from each other. Therefore, while the MI analysis used here provides useful insight into the strength of input-output coupling, interpreting it as a direct measure of encoding capacity overstates its significance. It is not clear that MI in this context extracts information beyond what could be obtained using conventional measures such as cross-correlation or coherence. To be clear, I’m not claiming the presented analyses are not insightful; I’m only disputing wording and interpretation. For example, the authors demonstrate that in PING-shifted models, PC output is more reliably modulated by theta input, whereas in ING-shifted models, Poissonian inputs are more effectively reflected in PC activity; the latter likely because ING models do not impose strict gamma-defined windows of opportunity for firing. I welcome clarification from the authors if my interpretations are incorrect.

Minor points

Regarding lines 115-120:

If I understand correctly, the authors attribute the difference in gamma frequency between θ-ING and θ-PING primarily to the relatively weak theta input. While I agree that a weak drive likely accentuates the difference by making PC firing sparser in θ-PING, I wonder whether the core reason might lie instead in the architecture of the circuits: in θ-ING, gamma is generated directly by fast-spiking BCs under theta drive, whereas in θ-PING it depends on the slower recruitment of BCs via pyramidal cell firing. Perhaps the authors could clarify whether this structural distinction, rather than input strength per se, is the main determinant of the frequency difference, or explain why that interpretation would be inaccurate.

Regarding Fig. 3:

I understand that the instantaneous firing rates shown in panels (v) are normalized to facilitate visualization of timing differences. However, it would help readers more fully grasp the results if some indication of the relative peak firing rates across conditions were also provided. For example, showing how much the overall firing rate of the pyramidal cell in the blue trace is reduced compared to that in the orange trace.

Reviewer #2: This manuscript integrates neural circuit dynamics modeling with experimental data analysis to explore how feedforward and feedback inhibition regulate the directionality of cross-frequency coupling in the theta-gamma band. The study reveals the mechanistic origins of cross-frequency directionality under different connections, and further validates its modeling predictions by incorporating in vivo electrophysiological data from the rat hippocampus. Overall, this research holds high scientific value and publication potential, though several minor revisions need to be addressed:

1.What do the different colors in the figures represent? It is recommended to add color legends or brief annotations for clarity.

2.Can similar gamma-to-theta phenomena be observed if the external theta input contains noise? The model should consider robustness under noisy conditions.

3.The model includes parallel pathways, while the experiments focus on a single channel. If the model is hypothetical, its assumptions should be further clarified, and the rationality of the model structure should be explained.

4.The experimental sample size is relatively small (5 rats). It is suggested to discuss limitations in the manuscript and propose hypotheses for generalizing the findings to other brain regions or non-human primates (e.g., monkeys) .

5.The calculation logic and intuitive meanings of CFD and CFC need more detailed explanation. These should be elaborated in the Methods section or Supplementary Materials. Additionally, since there is no illustration showing what "positive and negative values" represent in terms of temporal precedence, it is recommended to include a schematic diagram in the main text or Figure 1 to clarify that positive and negative CFD values correspond to "theta leads gamma" and "gamma leads theta," respectively.

6.Regarding computational modeling: As the study uses a simplified model, the simplification conditions and rationality of parameter selection need to be addressed—for example, through parameter sensitivity analysis and explicit justification of parameter choices.

Reviewer #3: In this manuscript the authors study theta-gamma cross frequency coupling as a function of structure of neuronal circuits. Namely they show that feed forward inhibition exhibit dominant gamma to theta interactions whereas there as the feedback inhibition mediated the reverse interactions. In addition, the authors also study the mixed circuit where they investigate different strengths of theta-BC drive and PV-BC coupling. They compared their results with the experimental data where rats explored familiar or novel environment.

This is a very interesting, well done study, with important implications on how the hippocampal circuit may function at different stages of information processing.

I have however few questions/comments that I would like the authors to answer:

1. the theta drive in the hippocampus is known to come from common source - medial septum. The authors implement dendritic delay on BC cells but not PC cells. why? Is it because PC compartments induce intrinsic delay? Where is the value of this delay taken from?

2. The conclusion drawn from experimental data seems a bit overdrawn. Based on Figure 3 similar results to those observed experimentally could be obtained just by strengthening BC->PC connection in ING circuit, rather than introducing PING circuit. This should be discussed

3. There is experimental data showing PV+ population mediates theta rhythm in hippocampus, and at the same is critical in consolidating newly acquired memory (see for example Ognjanovski et al, Nat. Comm 2017). What is experimental evidence for not including the theta drive to PV+ cells in the PING model?

4. Could authors explain why do changes in CFD organize vertically in the ING model (i.e are gamma frequency dependent; similarly to the data) where as they organize horizontally in the PING model (i.e. are theta frequency dependent)?

**Have the authors made all data and (if applicable) computational code underlying the findings in their manuscript fully available?**

Reviewer #1: Yes

Reviewer #2: None

Reviewer #3: Yes

PLOS authors have the option to publish the peer review history of their article (what does this mean? ). If published, this will include your full peer review and any attached files.

**Do you want your identity to be public for this peer review?** For information about this choice, including consent withdrawal, please see our Privacy Policy .

Reviewer #1: **Yes: ** Vitor Lopes-dos-Santos

Reviewer #2: No

Reviewer #3: No

**Figure resubmission:**
---

## [Decision Letter · Decision Letter 1]

25 Jul 2025

Dear Prof. Mirasso,

We are pleased to inform you that your manuscript 'The Role of Feedforward and Feedback Inhibition in Modulating Theta-Gamma Cross-Frequency Interactions in Neural Circuits' has been provisionally accepted for publication in PLOS Computational Biology.

Best regards,

Lyle J. Graham

Section Editor

PLOS Computational Biology

Reviewer's Responses to Questions

**Comments to the Authors:**

Reviewer #2: Authors have revised this paper carefully. Now, it can be accepted for publication.

**Have the authors made all data and (if applicable) computational code underlying the findings in their manuscript fully available?**

Reviewer #2: Yes

PLOS authors have the option to publish the peer review history of their article (what does this mean? ). If published, this will include your full peer review and any attached files.

**Do you want your identity to be public for this peer review?** For information about this choice, including consent withdrawal, please see our Privacy Policy .

Reviewer #2: No

---

## [Editor Report · Acceptance letter]

PCOMPBIOL-D-25-00474R1

The Role of Feedforward and Feedback Inhibition in Modulating Theta-Gamma Cross-Frequency Interactions in Neural Circuits

Dear Dr Mirasso,

I am pleased to inform you that your manuscript has been formally accepted for publication in PLOS Computational Biology. Your manuscript is now with our production department and you will be notified of the publication date in due course.

With kind regards,

Anita Estes
